# Comparison of Glucagon Stimulation Test and Insulin Tolerance Test for Assessment of an Anterior Pituitary Function—A Cross-Over Study

**DOI:** 10.3390/jcm14186567

**Published:** 2025-09-18

**Authors:** Krzysztof C. Lewandowski, Joanna Kawalec, Paulina Lewandowska, Wojciech Horzelski, Małgorzata Karbownik-Lewińska

**Affiliations:** 1Faculty of Medicine—Collegium Medicum, Mazovian University in Plock, Dąbrowskiego St. 2, 09-402 Plock, Poland; k.lewandowski@mazowiecka.edu.pl; 2Polish Mother’s Memorial Hospital Research Institute, Rzgowska St. 281/289, 93-338 Lodz, Poland; joanna.kawalec@onet.pl; 3Faculty of Medicine, The Medical University of Warsaw, Żwirki i Wigury St. 61, 02-091 Warsaw, Poland; plewandowska111@gmail.com; 4Faculty of Mathematics and Computer Science, University of Lodz, Stefana Banacha St. 22, 90-238 Lodz, Poland; whorzzelski@gmail.com; 5Department of Endocrinology & Metabolic Diseases, The Medical University of Lodz, Rzgowska St. 281/289, 93-338 Lodz, Poland

**Keywords:** insulin tolerance test, glucagon stimulation test, pituitary, dynamic tests, growth hormone, cortisol

## Abstract

**Background**: The insulin tolerance test (ITT) and glucagon stimulation test (GST) are commonly used for assessment of anterior pituitary function, but there are few direct comparisons of these tests, i.e., where both tests were performed on the same individuals. **Methods:** We designed a cross-over study, where we compared concentrations of glucose, cortisol, and GH during ITT and standard fixed-dose GST in 19 subjects (five males), with a mean age of 33.8 years (range 19–60) and a mean BMI of 27.8 kg/m^2^ (range 16.5–47.6). **Results**: Optimal fall in glucose concentrations during ITT (i.e., <40 mg/dL) was obtained in all study participants. During ITT we obtained lower minimal glucose concentrations (glucose nadir), i.e., 29.7 ± 7.67 mg/dL at 30 min of ITT, than during GST, i.e., 73.6 ± 9.67 mg/dL at 180 min of GST, *p* < 0.01. In contrast, glucose fluctuations (ΔGlucose) were higher during GST (77.8 ± 22.6 mg/dL versus 56.7 ± 10.9 mg/dL, *p* = 0.002, for GST and ITT, respectively). There was, however, no difference in degree of stimulation of either cortisol or GH release during both tests: ΔCortisol 9.28 ± 3.79 µg/dL for GST versus 8.49 ± 3.46 µg/dL for ITT, *p* = 0.4, and ΔGH 10.23 ± 10.36 ng/mL for GST versus 10.52 ± 9.67 ng/mL for ITT, *p* = 1.0. **Conclusions**: Although hypoglycaemia is not observed during GST in contrast to ITT, it appears that both tests lead to similar increments in cortisol and growth hormone secretion. We, therefore, conclude that GST should not be automatically considered as an “inferior” option in comparison to ITT.

## 1. Introduction

Due to diurnal rhythm as well as the pulsatile nature of anterior pituitary hormone secretion, formal assessment of hypothalamic–pituitary–adrenal (HPA) and somatotropic axes remains challenging. The insulin tolerance test (ITT) is used as the gold standard to evaluate the functioning of the HPA axis; however, there are many caveats associated with ITT. Given that biochemical hypoglycaemia (<40 mg/dL) is considered a prerequisite for a successful test [1], the test is very labour-intensive and requires constant medical and nursing supervision. This is because hypoglycaemia is thought to be the main factor stimulating sufficiently the secretion of either GH or cortisol [2,3,4]. Furthermore, ITT is not comfortable for patients and is contraindicated in individuals suffering from ischaemic heart disease or epilepsy. It is also not recommended to perform ITT in children and in the elderly [1]. Therefore, another test should be used alternatively, with the glucagon stimulation test (GST) being the most popular. GST has been extensively used in the UK to evaluate the functioning of the HPA axis and to evaluate the secretion of growth hormone (GH) [4,5,6,7]. Glycaemic fluctuations and, particularly, a fall in glucose concentrations after an initial rise, observed in the latter part of GST, are stimulatory factors for GH secretion [8]. In contrast, however, to ITT, such severe hypoglycaemia with glucose concentrations < 2.2 mmol/L (equal to 40 mg/dL) does not occur during GST.

There are, however, few direct comparisons of these tests, i.e., based on data obtained from investigations of the same patients. Hence, we have endeavoured to compare the performance of both ITT and GST in terms of the magnitude of stimulation of both GH and cortisol secretion.

## 2. Subjects and Methods

We compared concentrations of glucose, cortisol, and GH during ITT and GST in 19 subjects (five males), with a mean age of 33.8 years (range 19–60) and a mean BMI of 27.8 kg/m^2^ (range 16.5–47.6). Patients were investigated for amenorrhoea (n = 8), had a history of pituitary macro- (n = 2) or microadenomas (n = 5), isolated diabetes insipidus (n = 1), iatrogenic glucocorticoid-induced adrenal suppression (n = 1), hypopituitarism after congenital CMV infection (n = 1), or a history of cranial irradiation (astrocytoma) (n = 1). None of these patients received any hormonal treatment prior to the test, i.e., all were treatment-naïve for any replacement of an anterior pituitary hormone deficiency. Five of these individuals had a BMI of >30 kg/m^2^, while three individuals had a BMI of <20.0 kg/m^2^ (16.5 kg/m^2^, 18.7 kg/m^2^, and 19.6 kg/m^2^, respectively), seven had a BMI of 20.0–26.0 kg/m^2^, and four had a BMI of 26.0–30.0 kg/m^2^. There were no elderly individuals (the highest age was 60 years). We deliberately did not include very elderly subjects, as ITT is generally contraindicated in elderly individuals. All subjects had TSH and free T4 concentrations within the reference ranges and also had normal prolactin concentrations.

During GST, all subjects received glucagon intramuscularly (GlucaGen 1 mg HypoKit^®^; Novo Nordisk, Kalundborg, Denmark) in the morning after an overnight fast. The dose of glucagon was either 1 mg (for patients <90 kg) or 1.5 mg if the body mass was over 90 kg (fixed-dose protocol). Cortisol, glucose, insulin, and GH concentrations were subsequently assessed at 0–60–90–120–150–180 min after intramuscular glucagon. Though some authors suggest extension of GST up to 240 min, Leong K.S. et al. [4] demonstrated that extension of the test usually does not provide any additional data. Standard exclusion criteria to GST were employed, including diabetes mellitus and hyponatremia (defined as plasma sodium below 135 mmol/L).

On the following day, ITT was performed according to the standard protocol, where serum concentrations of GH, cortisol, and glucose were evaluated following intravenous injection of insulin (0.10–0.15 U/kg) at the time points of 0, 30, 45, 60, 90, and 120 min. Taking into account the recent data [9] that an insulin dose of 0.1 U/kg performed equally well as 0.15 U/kg during ITT, we administered insulin doses oscillating about 0.10–0.11 U/kg.

HPA axis integrity was considered complete if cortisol concentration at any time point during GST exceeded 375 nmol/L (equal to 13.6 µg/dL), according to Yo et al. [10]. The cutoff level for an adequate GH secretion during GST was defined as GH > 3 ng/mL [11]. There were, however, data [12] that such a cut-off point might be too high for obese individuals, so that other authors [13] suggested a lower concentration for GH (equal to 1.07 ng/mL) as a cut-off level during GST, particularly in obese individuals.

The magnitude of cortisol and GH secretion was calculated as the difference between the maximal stimulated cortisol and minimum cortisol concentrations (ΔCortisol), or between maximal (stimulated) and minimal GH concentrations (ΔGH) during either GST or ITT. It should be noted that in the case of GST, minimal cortisol concentration usually does not occur at time 0 min, as significantly lower concentrations are observed at 60 and 90 min of GST, prior to the peak observed that is usually at 150 or 180 min of the test [14].

GH concentrations were measured (in serum) by means of the immunochemiluminescence assay IMMULITE 2000 Xpi^®^ (Siemens, Munich, Germany). This assay has an interassay variation of 3% and an intraassay variation of 2.3%. Serum cortisol and insulin were assessed with an application of the electrochemiluminescence method (Elecsys Cortisol II on the Cobas 6000 platform; Roche, Basel, Switzerland). Interassay variation of this method is reported at 3.3% and intraassay variation at 2.6%.

The Ethics Committee of The Polish Mother’s Memorial Hospital–Research Institute in Lodz (Poland) has approved the protocol of this study (63/2020). In addition to the above ethical approval, all patients provided written permission prior to both tests and agreed that all their data and results of laboratory tests might be used for research. This remained in agreement with Regulation (EU) 2016/679 of the European Parliament and of the Council of 27 April 2016, where the above act pertains to the protection of natural persons with an adequate regard to the processing and free movement of personal data.

### Statistical Analyses

To analyse the data statistically, the MedCalc software, version 23, was used. A significance threshold of *p* < 0.05 was applied. Data distribution was assessed by means of the Shapiro–Wilk test. Spearman’s rank correlation coefficient was used to evaluate relationships between variables. For group comparisons, the Mann–Whitney U test was used for independent variables, while the Wilcoxon signed-rank test was applied for dependent (paired) variables.

## 3. Results

All results are presented in Table 1 and Table 2. All subjects in the ITT group achieved glucose concentrations below 40 mg/dL (2.2 mmol/L).

As expected, minimal glucose concentrations were lower during ITT (29.7 ± 7.67 mg/dL at 30 min of ITT) than during GST (73.6 ± 9.67 mg/dL at 180 min of GST, *p* < 0.01), although glucose fluctuations (ΔGlucose) were higher during GST (77.8 ± 22.6 mg/dL versus 56.7 ± 10.9 mg/dL, *p* = 0.002, for GST and ITT, respectively) (Table 1). There was, however, no difference in either cortisol (ΔCortisol 9.28 ± 3.79 µg/dL versus 8.49 ± 3.46 µg/dL, *p* = 0.4) or GH fluctuations during both tests (ΔGH 10.23 ± 10.36 ng/mL versus 10.52 ± 9.67 ng/mL, *p* = 1.0, for GST and ITT, respectively), Table 2.

Three patients in the GST group failed to reach cortisol concentrations above 13.6 µg/dL, and in two of these cases higher cortisol concentrations were observed during ITT (13.4 µg/dL versus 15.9 µg/dL (male, history of debulking of pituitary adenoma 70 × 40 × 35 mm accompanied by hypogonadism (LH < 0.1 IU/mL, FSH < 0.3 IU/mL, TSH 1.56 uIU/mL (0.27–4.2), FT4 0.88 ng/L (0.73–1.7), FT3 2.75 pg/mL (2–4.4)), morbid obesity (BMI 47.67 kg/m^2^), yet with maximal GH 1.21 ng/mL on GST versus GH 0.76 ng/mL on ITT). In the second case, maximal cortisol concentration during GST was 12.8 µg/dL versus 16.1 µg/dL on ITT (male, after two transsphenoidal surgeries for GH-producing adenoma, about 10 mm in size), while in the last case maximal cortisol concentration was 11.2 µg/dL on GST and 10.2 µg/dL in ITT (female, glucocorticoid-induced adrenal insufficiency).

On the other hand, in terms of GH secretion, in three cases we observed a “pass” on GST but an inadequate response during ITT (maximal GH 3.14 ng/mL on GST versus 2.42 ng/mL on ITT (female, secondary amenorrhoea, obesity, BMI 38.97 kg/m^2^, adrenal axis intact), GH 9.61 ng/mL on GST versus 2.48 ng/mL on ITT (male with isolated diabetes insipidus, adrenal axis intact, BMI 24.31 kg/m^2^), and GH 13.39 ng/mL versus GH 2.45 ng/mL (female, secondary amenorrhoea, obesity, BMI 39.3 kg/m^2^, adrenal axis intact). The opposite situation was only present in one case (maximal GH on GST 1.96 ng/mL versus 10.67 ng/mL on ITT—the case of the aforementioned man with a history of two transsphenoidal surgeries for GH-secreting pituitary adenoma four and three years prior to this admission, respectively). If an “old” cortisol cut-off of 18 µg/dL had been applied, then 6 out of 19 patients would fail to reach these cortisol concentrations on ITT (maximal cortisol concentrations of 15.9 µg/dL, 10.2 µg/dL, 15.9 µg/dL, 15.7 µg/dL, 15.3 µg/dL, and 12.3 µg/dL) and nine on GST (cortisol concentrations of 16.8 µg/dL, 13.0 µg/dL, 15.1 µg/dL, 11.2 µg/dL, 12.8 µg/dL, 13.8 µg/dL, 16.9 µg/dL, 15.8 µg/dL, and 17.5 µg/dL, respectively).

Hypogonadotropic hypogonadism was diagnosed in two cases (one pituitary adenoma, one case of cranial irradiation after astrocytoma). In women with amenorrhoea, final diagnoses were polycystic ovary syndrome (six cases—three of these obese with BMIs of 42.97 kg/m^2^, 39.33 kg/m^2^, and 38.97 kg/m^2^, and three with BMIs of 26.67 kg/m^2^, 21.56 kg/m^2^, and 26,13 kg/m^2^) and hypothalamic amenorrhoea (two cases). Regarding the last two cases, these included a woman with a history of eating disorders (BMI of 16.5 kg/m^2^, low LH of 0.5 IU/mL, FSH of 6.3 IU/mL, and oestradiol of 11.8 pg/mL) and another woman recovering from an eating disorder with a BMI of 20.32 kg/m^2^ who had normal gonadotropins (LH of 2.1 IU/mL and FSH of 4.4 IU/mL but still low oestradiol of 5.6 pg/mL).

## 4. Discussion

The current study is presumably the first study that directly compares GST and ITT in a cross-over design. It is widely recognised that formal evaluation of GH reserve requires dynamic testing. On the other hand, though the standard 250 µg short Synacthen test (SST) has been used for assessment of the HPA axis, it has the disadvantage of the application of a very high (i.e., supraphysiological) ACTH dose, and so there is a risk of false positive pass, as reported before [15].

In this context, ITT is considered to represent “the golden standard” for assessment of GH and ACTH/cortisol secretion. On the other hand, some authors claimed that GST can be considered as an effective test in the assessment of GH secretion, but it is inferior in terms of HPA axis assessment, i.e., in terms of the obtained magnitude of stimulation of ACTH and cortisol secretion [16]. It should be noted, however, that Berg et al. [16], while claiming a superiority of ITT in comparison to GST for HPA axis assessment, utilised a very high cortisol cut-off point of 599 nmol/L. On the other hand, other authors [5,17] demonstrated similar utility of both tests, i.e., both ITT and GST, in the assessment of an anterior pituitary function. Arch T. et al. also reported that mean maximal cortisol concentrations obtained during GST were not different in comparison to ITT [18]. Though our study did not include children, it is also worth mentioning that GST was found to be a reliable test of the hypothalamic–pituitary–adrenal (HPA) axis in children, though girls had higher peak cortisol at GST (mean difference of about 55 nmol/L), with even higher differences after the start of puberty (median of 672 nmol/L vs. 490 nmol/L, *p*  =  0.002) [19], though there are no data on whether such differences persist into adulthood.

In such settings, in our study we demonstrated that the magnitude of stimulation of both cortisol and GH (as assessed by ΔCortisol and ΔGH) is effectively identical both during GSH and ITT. Hence, both tests appear to perform equally well in the evaluation of either cortisol or GH secretion without any obvious “superiority” of one test over another. Clinicians must be, however, aware of the limitations of these tests (for instance, of the fact that GST does not perform well in people with diabetes). It should also be explained that the aim of this study was not to compare the number of individuals who pass ITT versus GST (for either peak cortisol or GH concentrations), but to mathematically compare the magnitude of either GH or cortisol release (i.e., ΔGH and ΔCortisol) after both forms of stimulation. This does not preclude a situation in which, in individual cases, one test might perform better than the other, as we have demonstrated that in three individuals (two of whom were significantly obese with BMI 38.97 kg/m^2^ and 39.3 kg/m^2^, respectively), GH concentrations were above 3 ng/mL on GST but fell between 1.07 ng/mL and 3 ng/mL on ITT. Notably, the opposite case (i.e., better stimulation of GH secretion on ITT than on GST) was observed in a man with morbid obesity (BMI 47.67 kg/m^2^).

Intramuscular glucagon is widely used for emergency treatment of hypoglycaemia with a half-life after intramuscular injection of only 26 min [20], so a one-day interval between GST and ITT was fully adequate for a complete wash-out.

With reference to the above, it is worth commenting on the cut-off level of cortisol applied in our study (377 nmol/L = 13.6 µg/dL). It should be noted that the cortisol cut-off point (500 nmol/L = 18 µg/dL) was originally established by Plumpton and Besser in 1969. These 500 nmol/L cut-offs were originally applied to ITT; however, they were also extrapolated into GST [21]. According to the current state of knowledge, this cut-off point is now considered too high. This is because currently used assays for cortisol measurements yield readings approximately 50 nmol/L lower than those used 50 years ago [22]. Therefore, and as suggested by Yo et al. [10], we decided to select the value of 377 nmol/L (13.6 µg/dL) as the cut-off level for cortisol concentration. The rationale for this was based on the fact that this cut-off level effectively predicts a 100% pass on the 250 µg short Synacthen test. The above-mentioned cut-off is also close to the updated ITT cortisol cut-off calculated at 416 nmol/L (15 µg/dL) as described by Lazarus et al. [22]. The issue of an optimal cortisol cut-off point for GST remains, however, open, as even a lower cut-off level for cortisol concentration (11.2 µg/dL = 310 nmol/L) during GST was established by Hamrahian et al. [23]. If this cut-off had been applied in our study, then all our subjects would have “passed” on GST. Nevertheless, we would recommend caution in the case of a woman with glucocorticoid-induced adrenal insufficiency who had an initial cortisol concentration of 4.8 µg/dL (0 min) and a minimal cortisol concentration of 3.3 µg/dL (30 min GST), though she demonstrated a subsequent increase in cortisol concentrations to 11.2 µg/dL. Notably, her maximal cortisol concentration was not higher during ITT (10.2 µg/dL). It should also be mentioned that in the case of GST, some patients with high initial cortisol concentration fail to produce further increase in cortisol on GST, as we have demonstrated before [24]. Nevertheless, high initial cortisol concentrations (that correspond to early morning cortisol, as both tests were started at 8 a.m.) have very high predictive values for the assessment of the HPA axis [25].

We need to stress, however, that although we had to accept certain cut-off points for subsequent clinical management, the aim of our study was not to clarify optimal cut-off levels for concentrations of GH and cortisol. Instead, we aimed to investigate and to compare the degree of both cortisol and GH release (i.e., ΔCortisol and ΔGH) after relevant stimulants. It should also be mentioned that both GST and ITT were performed in the same individuals on consecutive days, thus eliminating any possibility that the patient’s clinical condition might have changed during such a short time span.

Notably, a mechanism responsible for GH and cortisol secretion during GST is not fully known. Though glucose nadir was shown to contribute to GH secretion during GST [26], there was an attenuated but still significant increase in GH secretion even if a fall in glucose during the second part of GST was prevented by intravenous glucose infusion [27]. Notably, there was no relationship between insulin concentrations (maximal insulin concentrations, nadir insulin levels, or ΔInsulin during GST [26]) and fluctuations of either cortisol or GH observed during this test. Hence, mechanisms responsible for GH and ACTH-cortisol secretion during GST appear to be, at least partially, different [24] and, so far, have not been fully elucidated. Furthermore, while interpreting GH secretion after both tests, it is worth mentioning that in three of four discrepant cases (in terms of diagnosis of possible partial GH deficiency), GH concentrations were actually higher on GST than ITT, showing a clear “pass”. Non-inferiority of GST and ITT in terms of GH secretion was further confirmed by a complete lack of difference in ΔGH (10.52 ± 8.52 ng/mL (ITT) versus 10.23 ± 6.69 ng/mL, *p* = 1.0).

The main limitation of our study pertains to the relatively small number of participants. Nevertheless, results appear to be robust and quite convincing in the setting of a wide spectrum of endocrine conditions of investigated individuals.

In summary, we have demonstrated that both ITT and GST perform equally well in terms of the magnitude of stimulation of both cortisol and GH secretion. Hence, GST should not be considered as an “inferior” option in comparison to ITT.

## Figures and Tables

**Table 1 jcm-14-06567-t001:** Glucose, GH, and cortisol concentrations during ITT and GST (n = 19), a cross-over study.

	0 min	30 min	60 min	90 min	120 min	150 min	180 min
**Glucose ITT** **[mg/dL]**	86.1 * [86.0] ** ±7.03 #	29.7 [32.0] ±7.67	62.4 [62.0] ±13.62	72.9 [74.0] ±11.41	80.2 [81.0] ±10.48	-	-
**Glucose GST** **[mg/dL]**	86.6 [84.0] ±10.38	144.0 [144.5] ±23.11	133.2 [128.0] ±32.50	97.9 [88.0] ±31.01	82.3 [79.0] ±21.26	75.3 [77.5] ±14.21	73.6 [72.5] ±9.67
**Growth hormone ITT** **[ng/mL]**	0.99 [0.47] ±1.10	1.99 [0.96] ±2.57	11.10 [8.98] ±9.70	8.50 [4.69] ±8.45	4.06 [2.66] ±4.28	-	-
**Growth hormone GST** **[ng/mL]**	2.12 [1.24] ±2.36	1.32 [0.55] ±1.79	1.02 [0.50] ±1.18	3.19 [1.41] ±4.30	6.79 [1.9] ±11.01	8.05 [4.45] ±7.71	5.48 [3.07] ±6.30
**Cortisol ITT** **[µg/dL]**	15.8 [15.7] ±6.61	13.21 [11.9] ±5.84	19.0 [18.6] ±8.07	18.2 [16.5] ±8.77	15.6 [13.6] ±7.89	-	-
**Cortisol GST** **[µg/dL]**	14.8 [13.4] ±7.46	14.4 [14.1] ±7.09	13.2 [11.9] ±6.93	13.1 [10.7] ±9.21	13.1 [9.9] ±9.93	15.8 [14.5] ±9.43	17.2 [15.7] ±7.65

* Mean, ** Median, # Standard deviation.

**Table 2 jcm-14-06567-t002:** Comparison of differences between maximal and minimal concentrations (Δ) of glucose, growth hormone, and cortisol concentrations during the insulin tolerance test versus the glucagon stimulation test (n = 19).

	Insulin Tolerance Test	Glucagon Stimulation Test	*p*
**ΔGlucose [mg/dL]**	56.7 * [56.0] ** ± 10.9 #	77.8 [76.5] ± 22.6	0.002
**ΔGrowth hormone [ng/mL]**	10.52 [8.52] ± 9.67	10.23 [6.69] ± 10.36	1.0
**ΔCortisol [µg/dL]**	8.49 [7.9] ± 3.46	9.28 [8.0] ± 3.79	0.405

* Mean, ** Median, # Standard deviation.

## Data Availability

The original contributions presented in this study are included within this paper. Further inquiries can be directed to the corresponding author (malgorzata.karbownik-lewinska@umed.lodz.pl).

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
