# Peer review of "Comparison of Glucagon Stimulation Test and Insulin Tolerance Test for Assessment of an Anterior Pituitary Function—A Cross-Over Study"

_jcm, 2025, doi:10.3390/jcm14186567_

Round 1

Reviewer 1 Report

Comments and Suggestions for Authors

The sample size is very small (can not be representative)

In the results section, the authors did not mention how may patient were diagnosed with GH deficiency. They mentioned that 3 had high GH levels in GST versus ITT and 1 patient had high GH level in ITT versus GSH. So if no single patient had GH deficiency, so this number is not sufficient for comparison.

The authors mentioned a cut off level of GH < 3 ng/ml and one time they mentioned < 1 ng/ml. Which one they used ? and what is the reference for this very low cut off value

Comments on the Quality of English Language

Average

Author Response

The sample size is very small (can not be representative)

In the results section, the authors did not mention how may patient were diagnosed with GH deficiency. They mentioned that 3 had high GH levels in GST versus ITT and 1 patient had high GH level in ITT versus GSH. So if no single patient had GH deficiency, so this number is not sufficient for comparison.

We want to thank the reviewer for this comment. Though overall number of investigated individuals was small (n=19), the aim of the study was NOT to compare the number of individuals who pass ITT versus GST (for either peak cortisol, or peak GH concentrations), but to mathematically compare the magnitude of either GH, or cortisol release (i.e.  DGH and Dcortisol), after both forms of stimulation. Our data clearly demonstrate that in general there are no major differences in the degree of either GH, or cortisol stimulation between ITT and GST, though in individual cases one test might perform better than the other. We have made this clear in the discussion – lines 205-214. We also added reference nr 18 (Arch T et al. doi: 10.1507/endocrj.EJ18-0147), lines 193-194, where ITT and GST performed equally well in assessment of pituitary function, ands added referenced 6 and 7 to the Introduction.

The authors mentioned a cut off level of GH < 3 ng/ml and one time they mentioned < 1 ng/ml. Which one they used ? and what is the reference for this very low cut off value

We want to thank the reviewer for this remark. It was generally accepted that obese people have lower GH secretion on dynamic testing, so a lower cut-off point have been proposed, e.g. by Dichtel LE et al. J Clin Endocrinol Metab. 2014 Dec;99(12):4712-9. doi: 10.1210/jc.2014-2830. This reference has now been added (Ref 12). Please note, however, that the reference for the lower GH cut-off point was already in the text (reference nr 9, now reference nr 13].

Reviewer 2 Report

Comments and Suggestions for Authors

Comments to the Corresponding Author

In the paper titled "Comparison of Glucagon Stimulation Test And Insulin Tolerance Test For Assessment of An Anterior Pituitary Function – A Cross-Over Study" the authors aimed to determine if GST was inferior in comparison to ITT in evaluating cortisol and GH reserves.

The conducted study is designed as a cross-over study. The number of included patient was 19 adults (five males) mean age 33.8 years (range 19-60), mean BMI 27.8 kg/m2 (range 16.5-47.6).

The study results imply that the GST should not be automatically regarded as an ‘inferior’ alternative to ITT since both tests seem to produce comparable increases in cortisol and growth hormone secretion, although overt hypoglycemia is not typically seen during the GST, unlike in the ITT.

The paper is original, well-written and provides a comprehensive review of the relevant literature, however, the following clarifications are needed:

In the section 2. Subjects and Methods

  • Line 78. Please provide explanation why was ITT performed so soon (on the following day). Both the ITT and GST activate the hypothalamic–pituitary–adrenal (HPA) axis and GH secretion, which creates a period of refractoriness or exhaustion if they are performed too close to each other. It is necessary to allow full recovery of the HPA and somatotropic axes before repeating a stimulation test. In the literature and clinical practice: the minimum interval is 24–48 hours, but many endocrinologists choose 5–7 days so that the response remains reliably ‘fresh’ and adaptive mechanisms are avoided.
  • Please provide information about TSH, PRL, FSH and LH. This is important because of the following reasons:

TSH: In dynamic testing of the HPA axis (ITT and GST), endocrine practice requires that thyroid dysfunction be excluded or corrected beforehand. In primary or secondary hypothyroidism, baseline cortisol levels may be low and the stimulated response suboptimal. This can produce a false-positive diagnosis of adrenal insufficiency.

FSH and LH:  Eight patients were examined because of amenorrhoea. Evaluation of FSH, LH and estrogen would help in the differential diagnosis of hypogonadotropic hypogonadism or hypergonadotropic hypogonadism. Even though altered FSH and LH do not affect ITT or GST result, the relevant factor is low estrogen:

  • GH response to hypoglycemia or glucagon is often blunted.
  • Cortisol response is generally preserved, but interpretation of total cortisol must consider lower CBG levels.

PRL: Hyperprolactinemia itself does not directly influence the interpretation of ITT and GST results, but it is important to know if it is present. Dopamine is the main inhibitor of prolactin secretion in the pituitary gland but also plays an important role in the central regulation of the HPA axis and growth hormone (GH) secretion via the hypothalamus.
D2 receptors in the hypothalamus and pituitary participate in the regulation of CRH- ACTH-cortisol, and GHRH/somatostatin -GH. It is important to emphasize: Did the patients have hyperprolactinemia? Was it drug-induced? Were they receiving treatment for hyperprolactinemia? On the one hand dopamine agonists (bromocriptine, cabergoline) may blunt ACTH/GH response, and on the other dopamine antagonists (antipsychotics) may reduce stress responses due to altered hypothalamic regulation.

In the section 3. Results

Please provide results of glucose (fasting blood glucose, FBG), insulin and HOMA-IR which could be easily calculated from FBG and insulin.

Please provide BMI for all 19 patients.

In the section 4. Discussion

Please expand the discussion based on the added results about insulin resistance, obesity, thyroid status, estrogen level and presence of dopamine agonist therapy or dopamine antagonist cause of hyperprolactinemia.

Based on all the above, I recommend that the paper, should be considered for publication following revisions and additional clarifications.

Author Response

In the paper titled "Comparison of Glucagon Stimulation Test And Insulin Tolerance Test For Assessment of An Anterior Pituitary Function – A Cross-Over Study" the authors aimed to determine if GST was inferior in comparison to ITT in evaluating cortisol and GH reserves.

The conducted study is designed as a cross-over study. The number of included patient was 19 adults (five males) mean age 33.8 years (range 19-60), mean BMI 27.8 kg/m2 (range 16.5-47.6).

The study results imply that the GST should not be automatically regarded as an ‘inferior’ alternative to ITT since both tests seem to produce comparable increases in cortisol and growth hormone secretion, although overt hypoglycemia is not typically seen during the GST, unlike in the ITT.

The paper is original, well-written and provides a comprehensive review of the relevant literature, however, the following clarifications are needed:

In the section 2. Subjects and Methods

Line 78. Please provide explanation why was ITT performed so soon (on the following day). Both the ITT and GST activate the hypothalamic–pituitary–adrenal (HPA) axis and GH secretion, which creates a period of refractoriness or exhaustion if they are performed too close to each other. It is necessary to allow full recovery of the HPA and somatotropic axes before repeating a stimulation test. In the literature and clinical practice: the minimum interval is 24–48 hours, but many endocrinologists choose 5–7 days so that the response remains reliably ‘fresh’ and adaptive mechanisms are avoided.

We thank for this comment, yet, in our opinion, there is little evidence that would substantiate such a long-wash-out period. “Polish Mother’s” Memorial Hospital Research Institute is a tertiary referral centre for diagnosis of GH-deficiency in children with a catchment area of about five million population. In Poland testing for GH deficiency requires at least two dynamic tests to allow for growth hormone funding allocation. These tests are performed during a single hospital stay, and so experience of our paediatric colleagues clearly shows that 24hour interval is adequate between tests. Notably half-life of intramuscular glucagon is only 26 minutes (lines 215-217, see reference 20, now added). Half-life of intravenous insulin is even shorter (above 5 minutes). Furthermore, if the tests are conducted on consecutive days, then there is an advantage, that patients have virtually identical hormonal milieu. This is relevant, for instance in case of oestrogen concentrations, that vary several-fold during a menstrual cycle.

Please provide information about TSH, PRL, FSH and LH. This is important because of the following reasons:

TSH: In dynamic testing of the HPA axis (ITT and GST), endocrine practice requires that thyroid dysfunction be excluded or corrected beforehand. In primary or secondary hypothyroidism, baseline cortisol levels may be low and the stimulated response suboptimal. This can produce a false-positive diagnosis of adrenal insufficiency.

We want to thank the reviewer for this comment. We mention in line 76 and 77 that all patients had TSH, FT4 and prolactin within the reference range. This also included a patient with a history large pituitary adenoma – results quoted in lines 139-140.

FSH and LH:  Eight patients were examined because of amenorrhoea. Evaluation of FSH, LH and estrogen would help in the differential diagnosis of hypogonadotropic hypogonadism or hypergonadotropic hypogonadism. Even though altered FSH and LH do not affect ITT or GST result, the relevant factor is low estrogen:

We want to thank the reviewer for this suggestion. We now provide final diagnoses that was established in these women - lines 160-168, and in particular we also quote LH and FSH concentrations in women finally diagnosed with hypothalamic amenorrhoea.

GH response to hypoglycemia or glucagon is often blunted.

Cortisol response is generally preserved, but interpretation of total cortisol must consider lower CBG levels.

PRL: Hyperprolactinemia itself does not directly influence the interpretation of ITT and GST results, but it is important to know if it is present. Dopamine is the main inhibitor of prolactin secretion in the pituitary gland but also plays an important role in the central regulation of the HPA axis and growth hormone (GH) secretion via the hypothalamus.

D2 receptors in the hypothalamus and pituitary participate in the regulation of CRH- ACTH-cortisol, and GHRH/somatostatin -GH. It is important to emphasize: Did the patients have hyperprolactinemia? Was it drug-induced? Were they receiving treatment for hyperprolactinemia? On the one hand dopamine agonists (bromocriptine, cabergoline) may blunt ACTH/GH response, and on the other dopamine antagonists (antipsychotics) may reduce stress responses due to altered hypothalamic regulation.

We want to thank the reviewer for this observation. As mentioned above prolactin concentrations were normal in all subjects.

In the section 3. Results

Please provide results of glucose (fasting blood glucose, FBG), insulin and HOMA-IR which could be easily calculated from FBG and insulin.

Please provide BMI for all 19 patients.

We want to thank the reviewer for this suggestion. We provided these data in lines 71-74. We also provide these data regarding women with amenorrhoea lines 160-168.

In the section 4. Discussion

Please expand the discussion based on the added results about insulin resistance, obesity, thyroid status, estrogen level and presence of dopamine agonist therapy or dopamine antagonist cause of hyperprolactinemia.

We want to thank the reviewer for this valuable suggestion. Glucose concentrations are presented in Table 1. We did not measure in insulin in these subjects in view of the results of our previous study, where measured glucose and insulin at all time-points of GST in 139 individuals [Kawalec J, Horzelski W, Karbownik-Lewińska M, Lewiński A, Lewandowski KC. Determination of glucose cut-off points for optimal performance of glucagon stimulation test. Front Endocrinol (Lausanne). 2024 Aug 28;15:1448467. doi: 10.3389/fendo.2024.1448467. PMID: 39262672; PMCID: PMC11387979.]. We have previously demonstrated that there was no relationship, between either insulin peak, or a nadir, or D (delta) of insulin concentrations during GST and observed fluctuations in cortisol and GH concentrations. We now mention this in lines 254-256. We have, however, refrained from comments on possible effects of either dopamine agonists, or antagonist effects as well as oestrogen concentrations on the effects of dynamic pituitary testing, as this was NOT investigated in our study. We fully agree with the Reviever that secondary hypothyroidism may influence pituitary dynamic tests, yet thyroid axis is usually the last be affected, while ITT is generally not performed in patients with known deficiency of several hormonal axis, as diagnosis can be usually established without the need to resort to dynamic testing.

Based on all the above, I recommend that the paper, should be considered for publication following revisions and additional clarifications.

Reviewer 3 Report

Comments and Suggestions for Authors

This study cross-over study in which 19 subjects underwent both ITT and GST on consecutive days is interesting. The interventions and outcomes are well explained, with the standardization of the assays with a clear variability report. Suggestions to enhance this study include adding a cut-off justification. You use 13.6 ug/dl for cortisol, but to strengthen your results, the use of higher and lower doses as cuts off it will be a good justification.

Please also can add more details on clinical diagnosis as distribution,  hormone deficiencies, treatment history, that will be very useful to understand the variability between the patients.

The TT/GST was done on consecutive days. A short washout is probably fine, but some discussion of stress response carryover, like HPA priming, would be good to add. 

Highlight which patient groups (obese, elderly, post-surgery) GST may still be less reliable in the discussion.

Author Response

This study cross-over study in which 19 subjects underwent both ITT and GST on consecutive days is interesting. The interventions and outcomes are well explained, with the standardization of the assays with a clear variability report. Suggestions to enhance this study include adding a cut-off justification. You use 13.6 ug/dl for cortisol, but to strengthen your results, the use of higher and lower doses as cuts off it will be a good justification.

We want to thank the reviewer for this comment. The choice of cortisol cut-off is always debatable. We have however provided an information (lines 155-158), that six patients on ITT and nine patients on GST failed to reach 18 µg/dl cortisol cut-off.

Please also can add more details on clinical diagnosis as distribution,  hormone deficiencies, treatment history, that will be very useful to understand the variability between the patients.

We want to thank the reviewer for this valuable suggestion. Patients were treatment-naïve prior to investigations with an exception of the case of isolated diabetes insipidus (explained in lines 70-77). Hypogonadism was subsequently diagnosed in two males, while women with amenorrhoea had either PCOS, or hypothalamic defect (lines 161-168).

The ITT/GST was done on consecutive days. A short washout is probably fine, but some discussion of stress response carryover, like HPA priming, would be good to add.

We want to thank the reviewer for this observation. We provide information in lines 215-217 that intramuscular glucagon is widely used for emergency treatment of hypoglycaemia with half-life after intramuscular injection of only 26 minutes (see FDA Approved Drug Products: Glucagon Intravenous or Intramuscular Injection: Link: chrome-extension://efaidnbmnnnibpcajpcglclefindmkaj/https://www.accessdata.fda.gov/drugsatfda_docs/label/2015/201849s002lbl.pdf – now reference 20), so a one day interval between GST and ITT was fully adequate for a complete wash-out. Assuming possible HPA priming, the Dcortisol on ITT should be higher on ITT than on GST, that was, however, not the case. In such settings we concluded that discussion on HPA priming would be fully speculative.

Highlight which patient groups (obese, elderly, post-surgery) GST may still be less reliable in the discussion.

We want to thank the reviewer for this valuable suggestion. We have added information in lines 71-77 about BMI distribution in our patients. We also added that and there were no elderly individuals (the highest age was 60 years). We deliberately did not include very elderly subjects, as ITT is generally contraindicated in elderly individuals. We also added some information on good performance of GST in children (lines 194-199, reference nr 19 added).

Round 2

Reviewer 1 Report

Comments and Suggestions for Authors

still the problem of small sample size is a major issue